# Holonic Reengineering to Foster Sustainable Cyber-Physical Systems Design in Cognitive Manufacturing

Alejandro Martín-Gómez, María Jesús Ávila-Gutiérrez * and Francisco Aguayo-González

Design Engineering Department, Polytechnic School, University of Seville, 41011 Seville, Spain; ammartin@us.es (A.M.-G.); faguayo@us.es (F.A.-G.)
* Correspondence: mavila@us.es

**Featured Application: This paper presents the holonic model for evaluating and analyzing Cyber-Physical Systems for cognitive manufacturing systems. A technological mapping of the proposed holonic system based on the cyber-physical holon is presented.**

**Abstract:** Value chain is identified as the generator of the metabolic rift between nature and society. However, the sustainable value chain can mitigate and reverse this rift. In this paper, firstly, a review of the main digital enablers of Industry 4.0 and the current state of cognitive manufacturing is carried out. Secondly, Cyber-Physical Systems are conceived from the holonic paradigm, as an organizational enabler for the whole of enablers. Thirdly, the bijective relationship between holonic paradigm and container-based technology is analyzed. This technology allows mapping the physical and virtual holon as an intelligent agent embodied at the edge, fog and cloud level, with physical and virtual part. Finally, the proposed holonic system based on the cyber-physical holon is developed through multi-agent systems based on container technology. The proposed system allows to model the metabolism of manufacturing systems, from a cell manufacturing to whole value chain, in order to develop, evolve and improve the sustainable value chain.

**Keywords:** sustainable supply chain; sustainable value chain; circular economy; holonic systems; container technology; multi-agent system; Cyber-Physical System; enablers; Industry 4.0

## 1. Introduction

Regarding the digital transformation strategies of the value chain, many factors must be considered, including population demographics and available skills. In addition to complexity, the global digital revolution is taking place on multiple levels simultaneously. The lowest level relates to digital tools and real-time connectivity that enable interaction and integration performance between people and machinery. The second level focuses on digital platforms and digital markets that connect industries through digital value chains and interrupt old markets and business relationships. Europe has strong digital assets, but emerging platform markets are dominated by US and Asian players. The third level relates to the global supply of expertise and resources, which is based on machine and platform levels.

The inputs contained in this paper are valuable to the implementation of Europe's Industrial Strategy for Digitalization. Digital Innovation Hubs (DIH) catalogue has been created to contain comprehensive information on Digital Innovation Hubs in Europe to assist with networking among DIH across Europe [1]. This is a technology for the early stages of a reengineering project that integrates the resources of regional and European DIH networks through Cyber-Physical Production System (CPPS) technologies for Industry 4.0 ecosystems, with networks such as FORTISSIMO and BeinCPPS. These networks have been instrumental in making Cyber-Physical Systems (CPS) and high performance computing technologies available to Small and Medium Enterprises (SMEs) and how they have exploited such computing capabilities to build significant competitive advantage.

Europe's efforts towards DIH and digital platforms aim to increase regional competitiveness and prosperity, while maintaining a strong focus on climate challenges and a sustainable European labour market [1].

Until now, the value chain has been identified as the origin of the metabolic rift between society and nature, and at the same time as a means to mitigate and reverse it [2]. In this context, the value is conceived and modeled to allow its aggregation throughout the chain, not only in the traditional economic vector, but also in the environmental and social vectors, that is, from the three pillars of sustainability. Therefore, the circular value chain, from the appropriation model of natural resources, represents the level of analysis with the highest degree of aggregation of the productive system, so it is suitable as an object of implementation of the circular industrial metabolism.

Thus, digital transformation is an aspect that is acquiring a special interest for the circular value chain that, from the principles of waste disposal of the lean philosophy in the value chain, must use organizational and digital enablers of Industry 4.0 to evolve towards the lean digitalized circular value chain.

The complexity of manufacturing environments, of socio-economic and natural environments, which are constantly evolving, must be managed by an enabler who has the ability to adapt to the environment and evolutionarily implement the necessary mechanisms that enable the integration of the organization. The holonic is presented as an appropriate organizational enabler for the variety required from the complexity of the market manufacturing ecosystems and natural ecosystems.

Therefore, the opportunity for digital transformation from value chain 3.0 to value chain 4.0 is formulated. Thus, value chain is modeled as CPS (and digital twins). This is carried out under the organizational enabler of the holonic model. This holonic model, multilevel and multiscale, enables the management of the integrated value chain in the territory, taking into account territorial axes and vertical and horizontal synergies. From the opportunities of Industry 4.0, specifically from the CPS, it is proposed the development of a virtual entity, or virtual-digital twin, of each physical entity (product, process, machine, manufacturing plant, etc.), which would allow to evaluate dynamic changes taking into account the constant flow of data. This system aims to answer questions regarding social, economic and environmental aspects. Likewise, another important aspect is the optimization in manufacturing systems (not only at economic level). In this field, the incorporation of surrogated models in industrial sector from circular economy allow manage the complexity of the system in a lighted way [3].

According to the conception of the value chain as a CPS in which the productive elements are hybridized as a digital twin carrying data and information, the opportunity to operate on them from cognitive computing and artificial intelligence presents itself. The above must be considered simultaneously with the possibility of embedding intelligence in the physical dimension that constitute the CPS, enabling the dimension of cognitive computing not only for optimization and local control, but for the intention with humans in the natural way, improving social sustainability. A natural extension of cognitive computing and artificial intelligence is its application to carry out the transition from a linear to a circular model of appropriation from the paradigm of the circular economy and the pillars of sustainability [4], to mitigate the metabolic rift that has originated the transformation of the value chain by separating the social and natural dimension [5]. The above requires frameworks to carry out the reengineering of the existing value chain under the principles of the circular economy, among which are solutions based on natural systems, considering nature as a model, measure and mentor. Among the different frameworks that are oriented in this direction, there are fractal, bionic and holonic manufacturing. In this paper a framework for the reengineering of value chains as holonic CPS is presented.

Holonic manufacturing systems (HMS) are currently some of the most studied and referenced solutions for the next generation of manufacturing systems; these solutions provide the necessary guidelines to create open, flexible and agile control environments for the smart, digital and networked factory [6].

One of the strengths of the holonic paradigm is that it enables the construction of complex systems exhibiting hierarchical behavior, highly resilient to disturbance, and adaptable to changes in the environment [7].

After around thirty years of use of the holonic paradigm, an evaluation of the relevance of this technology related to reengineering of manufacturing process is presented in this paper. The main contribution in this paper in relation to the state of knowledge in Holonic manufacturing is the realisation of a differentiated architecture based on a fractal structure of smart product, smart process and smart facility different from other holonic architectures (e.g. PROSA or ADACOR architectures) as sustainable Cyber-Physical Systems that constitute a disruptive innovation in manufacturing systems and its conceptual mapping through the arrowhead microservices architecture and its implementation with container-based information technologies.

This paper is organized as follows: the Background section provides the main enablers from Industry 4.0 and concepts developed in order to support the reengineering of cognitive manufacturing systems. The main section describes the conceptual framework and its requirements for develop the cyber-physical holon and the way to implement it through the container-based technology. Finally, last section presents the conclusions.

## 2. Background of the Literature

In this section, a review of the main concepts related to this work is carried out as shown in Figure 1. This section describes the concept of cognitive manufacturing, the proposed architectures for its implementation, for projection into the proposed framework of Holonic reengineering of the sustainable value chain. Afterwards, it proceeds to establish the current status of the main enablers coming from Industry 4.0, which enable the necessary evolution of cognitive manufacturing. The presentation is carried out on the domains in which the intelligence is embedded based on the characteristics of information and communication technology (ICT) infrastructures and services, structured in cloud, fog and edge computing.

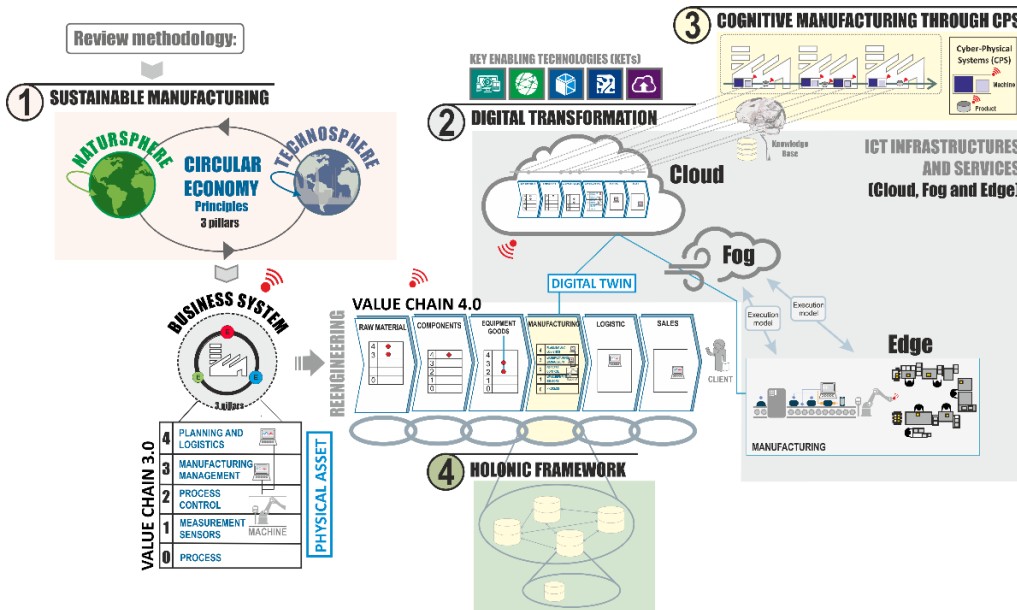

**Figure 1.** Research methodology.

### 2.1. Sustainable Manufacturing

The development of manufacturing systems has led to an imbalance in the situation that initially existed between humans and nature. This disruption is known as the metabolic rift [8,9] or the rupture of the natural connection between the flows of matter and energy that initially existed between labour, society and the natural environment. This has motivated

research into how to carry out direct engineering and reengineering in manufacturing systems that try to mitigate this separation from the points of view of sustainability, by being conceived as an analogy to natural systems [10–12] and incorporating the potential of the most modern information technologies or enabling technologies [13–15] that allow obtaining intelligent, connected and sustainable manufacturing systems.

Several definitions of sustainable manufacturing (SM) have been proposed to date. Among them, the US Department of Commerce (DOC) [16] defines sustainable manufacturing as "*the creation of products that use processes that have the least negative impact on the environment, conserve energy and natural resources, are safe for employees, communities and consumers, and are economically robust*".

In terms of research on the concept of sustainable manufacturing, publications can be found associated with impacts related to energy consumption [17], water consumption [18], use of materials and substances and waste [19,20]. In regards to product recovery and environmental awareness are the works of Ilgin et al. [21] and Krill et al. [22] for remanufacturing processes, Ramani et al. [23] for sustainable life cycle design and finally at the micro manufacturing level there are quality of sustainable manufacturing processes [24,25], emissions [26] improved design and machining [27–33]. For a more extensive view of sustainable manufacturing, the contributions of Gunasekaran and Spalanzani [34] are worth considering. They divide sustainable manufacturing into seven general fields that include the identification of problems and opportunities for sustainability in manufacturing systems, the supply chain and services, organisation and manufacturing and design, that is, they structure their review at the different macro-meso and micro levels of production systems. On the other hand, other interesting reviews include those by Haapala et al. [35], Young et al. [36], Westkämper [37], Fratila et al. [38] and Depeisse [39], among others, who are among the most representative authors on the topic of sustainable manufacturing.

There is a large number of researches that has attempted to partially naturify manufacturing systems with a bio-inspired approach in order to develop and implement technical systems in natural systems in an eco-compatible way. The scope of these research works across the entire value chain of manufacturing systems at different levels such as: in the supply chain and virtual organisations [40], in which bio-inspired oriented artificial intelligence techniques are used such as: behavioural-based algorithms in order to optimise machining times [41], adaptive manufacturing coordination and control systems [42] or the use of bee-based algorithms for manufacturing cell optimisation [43], optimal task allocation [44] and routing and scheduling of manufacturing plants [45].

Nowadays sustainable engineering has a number of challenges and opportunities in order to configure an integrated and eco-compatible metabolism between the technosphere and the natursphere. For this reason, it is necessary to consider in the present and future projects an increase in their complexity, and the need for the development of sustainable technologies and systems that give support in these new projects, with the Sustainable Development Goals (SDG) under the 2030 Agenda [46].

Until now, no unified multi-scale and multi-level paradigm has been proposed that integrates, under criteria of sustainability, minimum complexity and required variety, the natursphere with the technosphere, mitigating or reversing the metabolic rift by conceiving manufacturing systems with an eco-compatible variety with nature.

## 2.2. ICT Infrastructures and Services

In the context of reengineering the value chain of manufacturing systems as a successful enabler is the existence of information and communication technology infrastructures under 4G and 5G technologies that enable connectivity, bandwidth, and latency, and that has determined the conception of an information and communication systems infrastructure under an architecture of resources organised in three levels: (1) Cloud, (2) Fog, and (3) Edge, incorporating the full potential of the technological enablers, to conceive manufacturing systems as second nature that mitigates the metabolic rift through the naturification of technical systems. Below is a brief synthesis of how ICT infrastructure and enablers

are being incorporated into the direct engineering and reengineering processes of the sustainable value chain and its manufacturing systems.

### 2.2.1. Cloud Manufacturing

Regarding cloud computing, the concept of cloud manufacturing (CMfg) is presented as an innovative manufacturing paradigm [47], developed based on advanced manufacturing models, IoT, business information technology, service-oriented technologies and virtualization [48]. From this point, the interest in the projection of cloud and CPS in Industry 4.0 is determined [49,50].

CMfg is understood as a model to enable ubiquitous, convenient, and on-demand access by the network to a shared set of configurable manufacturing resources (e.g. manufacturing software tools, manufacturing equipment, and manufacturing capabilities) that can be quickly provisioned launched with minimal management effort or service provider interaction [51]. In accordance with this approach, CMfg is a concept intended to offer on-demand manufacturing services for networked manufacturing resources (that is, enabled for the cloud). An example of this is design as a service, simulation as a service, production as a service, assembly as a service, testing as a service and logistics as a service.

Since manufacturing resources and capabilities are shared (as services) over the Internet, CMfg, in particular, is considered beneficial to small and medium-sized enterprises [52]. In this context, manufacturing resources and capabilities are virtualized and organized into a group of resources, therefore, all CMfg partners can perform manufacturing tasks in real time and thus enabling collaborative environments [53].

The core idea of CMfg is to connect and integrate the manufacturing resources of different factories (or companies) in the cloud to enable resource sharing and collaboration on a large scale, including the end user in the process [54]. Along these lines, in recent years various approaches have been proposed to include virtualized manufacturing resources, so that they can be cloud-based manufacturing services [47,55–57]. The use of CMfg in the supply chain can also have an impact on the integration of chain information, and on physical and economic flows [58].

### 2.2.2. Fog Computing

Fog computing is a term that emerged to support the requirements of IoT applications that today's solutions could not meet. Cloud integration of IoT applications is not easy to manage, especially due to latency issues. On the one hand, it provides substantial advantages for both providers and end users, on the other hand, it poses a new unsustainability in the integration with ubiquitous services. Although the cloud can lead to huge improvements in system processes thanks to its vast number of available resources, direct exploitation of cloud resources by ubiquitous IoT devices can present several technical challenges and inefficiencies. Among these challenges are network latency, traffic and communication overhead to devices, and of course the costs of connecting a large number of sensors directly to the cloud, this being extremely demanding on cloud resources. [59,60]. The result is that the cloud remains occupied for each sensor duty cycle, and therefore the bandwidth cannot support this data load. In this sense, fog computing provides computing improvements, decreased latency periods, also allowing load balancing on various servers, privacy and security improvements, accessibility, affordability, feasibility and maintenance [61].

The composition of fog computing topology provide real-time integrated machine learning using cyber interactions [62,63]. The implementation of fog computing involves making decisions about the implementation of fog hardware in terms of different options, such as: Fog, micro and container data center; In addition, it is necessary to determine the processing needs and connectivity of fog with edge and cloud.

Fog computing, has its initial origin in the way to solve some of the limitations of ICT infrastructure, such as bandwidth and latency in the current 5G era, through close proximity to the edge infrastructures that guarantee bandwidth and latency by providing

real-time support for management and optimisation needs. These functions will be what will make Fog infrastructures survive under 5G.

### 2.2.3. Edge Computing

The edge level of the multilayer architecture of industrial computing systems 4.0 is located in the field, where the productive equipment and the operational flow are located. At this level, data is obtained from sensors and PLCs. Based on the captured data, the real-time processing of the data is carried out with intelligence on board the equipment, establishing optimized operational control strategies, which determines the signals to send to the actuators and monitoring for supervision and control by human operators. In addition, edge communicates with the fog computing layer where the stored data arrives. Processing strategies are established for optimization with broader objectives such as at the process or department level with the intelligence supported by the microdata-centers that constitute them.

The edge has been made possible thanks to the possibilities of M2M, M2P connectivity, with special emphasis on wireless personal connections in industrial plants, intelligent computing hardware and software embedded in devices, equipment and the incorporation of IoT [64].

The inclusion of edge computing for the set of operational functions and tasks in operation of Industry 4.0 entails a set of benefits [65], among which are: (a) data capture and processing for control, optimization and local monitoring at the machinery level with the latest developments in manufacturing processes and optimisation techniques in key sectors such as aeronautics [66]; (b) optimization by team cooperation within a process; (c) decreased vulnerability of Industrial Systems by displacing cloud and fog services to edge computing, improving security and quality; (d) reduction of the latency required to obtain the necessary response in the operational system, as Cloud resources are required on a delayed basis; (e) decrease in bandwidth requirements; (f) Hot scalability from the needs of the field in which the industrial machinery is located; and (g) reduction of costs to require less time of the cloud resources that are paid for the services consumed.

At the edge level, the surrogate models from the cloud and fog are housed to establish the operational parameters of the machinery from which new data is obtained that allows control and optimization in real time-synchronous-by embedded local intelligence. These data are also sent to fog and cloud, where they can be processed with Big data techniques to perform in the cyber-physical dimension of industrial machinery. This is through simulations driven by data over time-diachronic-developing new surrogate models that will be periodically sent to fog and edge from the cloud.

Edge reengineering in the digitization process under Industry 4.0 involves making edge hardware, edge software and edge connectivity decisions.

The ecosystem of CPS has a real part and a virtual part for whose implementation edge computing systems are required, which sensorize and date the real dimension of objects and processes [67]. Next, edge enablers that allow the design and implementation of the physical dimension of CPS are identified.

Beacons use a low consumption Bluetooth communication, its main function is to send positioning signals [68]. The unique beacon identifier can be associated with certain information on a cloud server, enabling smart services or IoT [69]. Radio frequency identification is a system capable of storing a large amount of data through devices called RFID tags [70]. It has the ability to measure environmental factors such as temperature, humidity or pressure [71]. NFC is a more recent version of RFID, currently being integrated with mobile devices in a bidirectional communication between the NFC tags and reader [72]. At the manufacturing level, mobile applications establish real-time communications with machines [73], CPS [74], as well as with processes and between workers [75–78].

At the communication level, machine-to-machine (M2M) is a new communication technology through which a large number of intelligent devices can communicate autonomously and make collaborative decisions without direct human intervention [79]. It

makes it possible to capture data, coordinate CPS and deploy remote services, ensuring that everything is in real time and ubiquitously [80]. This automatically increases the efficiency of the products. M2M communication finds applications in areas such as smart grids [81], LAN, health, intelligent transport systems, environmental monitoring and manufacturing industry, among others [82]. At the computational level, wearable technology encapsulates a large number of devices intended to be used by and on people [83,84]. Wearables gain importance by converting physical elements into digital information for further processing [85]. In addition, they can benefit workers in terms of efficiency, productivity and safety by adding value to manufacturing processes. Likewise, Middleware is an important facilitator that provides communication between heterogeneous devices. It is an intermediate layer between devices and application services and provides an abstraction of device functionality for application services [86–88].

Within this field of edge computing, it is also worth noting the ubiquitous computing in the manufacturing sector, which represents the paradigm of designing, manufacturing, and selling anywhere, anytime [89,90]. Embedded computing, which consists of incorporating sensors and low-cost intelligent computing equipment into machines, with communication possibilities with other intelligent objects and machines [88]. As well as the concept of intelligent environment [89], which consists of the integration of sensors with intelligence and connectivity in the production and logistics elements, in the operational environment and in the facilities.

### 2.3. Cognitive Manufacturing through CPS

In this section, cognitive manufacturing systems will be addressed together with one of the main technological enablers of Industry 4.0, which are CPS formed by the hybridisation of physical manufacturing resources and their virtual digital twin in the cloud. For this purpose, the section will be divided into: (1) Cyber-Physical Systems (CPS); (2) subrogate models and (3) cognitive manufacturing, which is the synergetic result of the previous elements and the potential of cloud, fog and edge computing.

### 2.3.1. Cyber-Physical Systems (CPS)

CPS are the main enabling technology for Industry 4.0. These CPS allow objects and processes residing in the physical world (e.g. facilities in a manufacturing system) to be coupled and evaluated using advanced predictive analysis (e.g. machine learning models) and simulation models in the cyber world, with the intention of auto-configuring operations.

CPS enables objects and processes in the physical world to be closely related to computing, communication and control systems in the cyber world [90]. Cyber-physical interfaces connecting both worlds enables transmissions using wireless sensors, smartphones, and tablets, among others [91]. Conceptually, these cyber-physical interfaces present cyber-digital twins, where real-world physical objects are represented as virtual objects in the cyber-world. In turn, these virtual objects can be analyzed, interrogated or simulated individually and/or collectively to derive operational knowledge and inform for decision-making. That enables the creation of the CPS ecosystem.

In this area, IoT is presented as a fundamental enabler since it includes devices with Internet access and gateways to detect, collect, send and receive data [92]. In terms of manufacturing, this may involve interactions with sensors, controllers, actuators, RFID tags, GPS, and high-definition cameras [92], among others. Obviously, these ongoing interactions describing factory operations produce large repositories of data [90]. Once enough data of adequate quality has been captured, these data sets can be analyzed using machine learning to make predictions (e.g. equipment failures). It proposes to structure IoT-based Cyber-Physical Systems by means of holons, communicating modular units, arranged in part-whole hierarchies, which host a behaviour, accessible through an interface, whereby the holon plays recursively and at the same time the double and complementary role of part and whole. Other approaches propose structuring CPS based on the IoT through holons, communicating modular units, arranged in part-whole hierarchies, which

host behaviour, accessible through an interface, whereby the holon recursively plays the dual and complementary role of part and whole at the same time [93].

The fast development of CPS makes it necessary to establish analysis frameworks to classify them. Cardin [94] proposes a classification framework based on seven criteria: degree of development (laboratory, learning, industry); research axis (agility, technology and sustainability); instrumentation; communication standards; intelligence repository; level of cognition (degree of developmental maturity with respect to cognition); and human factor.

### 2.3.2. Surrogate Models

Traditionally, modeling is done with the help of mathematical models that describe physical processes and phenomena that occur during the operation of an object using complex differential equations with boundary conditions. These equations are solved using complicated numerical methods that require significant computing resources and a lot of effort in preparing input data. Consequently, in recent years, emphasis has been placed on data-based mathematical models using the results of large-scale and computational experiments. In other words, the models are trained on a set of input and output data prototypes and simulate (replace) data sources based on an initial model and the models created. These adaptive models are sometimes also called metamodels (models on models) or surrogate models [95]. The methods and techniques used to build these data-driven models take advantage of the synergy of general scientific disciplines, such as mathematics, AI, data analysis, visual computing and IoT [96] and ICT. Examples of these models can be found in the field of CAD modeling [97], power generation systems [98,99] and power consumption [100], models for the optimization of solutions in the design of low-energy buildings [101], or geochemical simulation models driven by data [102]. Big data and AI techniques allow one to build surrogate models to improve simulations and reduce model calibration and run times [103].

Optimization has evolved considerably over the years, incorporating more recently the evolutionary computing assisted by subrogated models in circular economy in industrial sectors [3]. Although, beyond subrogation models based on regressions, neural networks, stochastic models, etc. It is possible to conceive the subrogated models as lightened models of the cognitive models (cognitive computation) that are present in each of the levels of the value chain in the manufacturing systems [104]. Industry 4.0 and its enablers are the ones that provide adequate support for the development and exploitation of these subrogated models. These models could contemplate more than one dimension, thus managing the complexity of the system, housing the required variety.

From the opportunities of Industry 4.0, specifically from the CPS, the development of a virtual entity, or virtual-digital twin, of each physical entity (product, process, machine, manufacturing plant, etc.) is proposed, which would allow evaluating dynamic changes taking into account the constant flow of data. This system aims to answer questions regarding the social, economic and environmental aspects.

### 2.3.3. Cognitive Manufacturing

Cognitive computing in terms of manufacturing denotes that machines and processes are equipped with cognitive abilities [105]. In technical terms, this includes sensors and actuators that allow machines and processes to evaluate and increase their operating range autonomously. Knowledge and learning models equip the factory with information on their capabilities, thus helping to expand the capabilities of machines and processes. When performing its tasks, the manufacturing environment acquires models of the manufacturing processes, machine capabilities, parts and work tools, their properties, as well as the relevant contexts of the manufacturing processes. They differ from other technical systems in that they perform cognitive control and have cognitive abilities such as perception, reasoning, learning and planning, with a specific architecture.

Particularly, cognitive technical systems are systems that know what they are doing and should be able to [106]: reasoning from knowledge models, planning their own actions, learning from experience and instructions, responding firmly and astutely to surprises, explaining and be aware of themselves, and adapting to humans.

In this area of cognitive manufacturing, there are frameworks for autonomous design and manufacturing, manufacturing planning [107], provider of cloud manufacturing solutions [108,109], among others. Currently, there are various architectures of cognitive manufacturing systems that contemplate the potential of Industry 4.0 [110], where are identified numerous examples that use cognitive computing along with Big data [111], such as the case of IBM Watson [112], cloud [113], edge computing and internet of things (IoT) [114]. Thus, the architecture of iRobot Factory [115], contemplates the perspectives of intelligent terminal, systems administration, edge computing, cloud computing, cognitive computing, intelligent device units and the production line layer of the industrial manufacturing plant.

Aligned with the CPS, the advantages of Industry 4.0 and the study of human-machine interaction, joint cognitive systems (JCS) appear. These systems are characterized by three aspects or principles [116]: (a) goal orientation, (b) control and (c) co-agency. The first principle states that all agents are goal-oriented; the second aspect refers to the principle of working together to improve control and minimize entropy (e.g. disorder in the system), and the third aspect is about the interdependent and interrelated nature of the actions of all agents within a JCS.

### 2.4. Holonic Paradigm

One of the aspects that characterise the evolution of manufacturing systems is the open and continuous innovation coming from very diverse technologies, and the necessary mitigation of the environmental and social load, with a strategy to reduce the intensity of biological nutrients on the natursphere and technical nutrients on the technosphere, as well as their complexity. This determines the need for a long term support (LTS) architecture that offers the necessary solutions for a longer period than normal, taking into account aspects of continuous innovation, required variety and eco-compatibility, multilevel and multi-scale complexity, so as to ensure their adaptive nature and co-evolution. In response to these requirements, Holonic architecture can be found.

The holonic paradigm is based on the work of Arthur Koestler, who formulated a model of the structure and behaviour of complex systems, considering them as made up of entities that are both whole and part [117]. To describe a basic unit of organisation in social and biological systems, Koestler created the word "holon", which comes from the combination of the Greek word "holos" (whole), and the suffix "on" (part). In the domain of a social organisation a holon acts as a part of a whole and as a whole for its parts, depending on the perspective adopted. Koestler also proposed the concept of Open Edges Hierarchy (OEH) as the architecture formed by holons, called holarchy, which is not bounded either downwards or upwards in its structure [118].

The first applications of holonics in manufacturing came from Japanese researchers in the 1980s with the design and implementation of a holonic controller for a manipulator. Hirose et al. [119] presented this design and in a later work [120] described the software for it. The prototype implementation for the manipulator was presented in 1990 [121]. The advantages described by the application of the holonic paradigm in this development were a more robust design due to reduced wiring and increased reliability of the manipulator. The manipulator control software required coordination between the internal controllers, through the use of specific task managers and message exchange, which is common in holonic control software.

Using the holonic paradigm in the design of manufacturing systems appeared in the early 1990s in the Intelligent Manufacturing Systems (IMS) programme as a solution to the increasing rate of change affecting the business world in general, and indeed the manufacturing sector as well [122]. A consortium of researchers from Australia, Canada,

Europe, Japan and the USA was created to develop the tools and implementation of a holonic framework in real-world industry to realise the potential benefits offered by holonic organisations, such as "*stability in the face of turbulent and chaotic environments, adaptability to cope with change, and efficient use of available resources*" [123]. To guide researchers in this area, participants in the IMS consortium established a set of definitions as a framework for the constituent entities of holonic systems [124], which evolved into the Holonic Manufacturing Systems (HMS) [125].

HMS are presented as a potent approach to develop smart connected manufacturing systems. HMS is conceived as a distributed manufacturing system in which each component, element and/or manufacturing resource is modelled and controlled by holons that can cooperate in solving complex problems, a holon being an autonomous and flexible computational entity capable of social interaction, and the ability to communicate and cooperate with other holons [126].

The literature contains a wide range of interpretations of Koestler's concepts. In the field of manufacturing systems, at the micro level, several architectures have been proposed including: Product Resource Order Staff Order Architecture (PROSA) [123]; Adaptive Architecture for the Control of Manufacturing Systems (ADACOR) [127]; Holonic Component-Based Architecture (HCBA) [128]; Holonic Unit of Production [129]; META-MORPH [130].

In addition to the mentioned architectures, there are other architectures such as: Holonic Control Device (HDC) [131], Fabricare [132], ANEMONA [7], Holonic Shopfloor Control System (HSCS) [133], Concurrent Integrated Process Planning System (CIPPS) [134] and Product, Resource, Order, Simulation Isoarchic Systems (PROSIS) [135].

Based on holon theory, there are research groups, such as the Xi'an Engineering College of the Armed Police Force (China), that propose using the holonic structure for product control in distributed manufacturing environments [136–139]. These authors suggest the advantages of using a holonic structure, because the holonic structure possesses a number of characteristics that are similar to those needed in distributed enterprises. Examples of these qualities are that holons are distributed, decentralised, autonomous, dynamic, reactive, flexible and, adaptable, all of these qualities contribute to improve the control of product manufacturing.

The review carried out shows that HMS have traditionally been oriented towards optimising manufacturing, control and planning systems, being aimed at providing high efficiency from an economic perspective, and not taking into account the other two perspectives of sustainability (social and environmental).

Research into holonic manufacturing control architectures has identified approaches and models that aim to incorporate strategies and mechanisms that guarantee the robustness of the system in the face of unexpected changes. This makes it possible to improve the resilient behaviour of the system, an aspect found in nature from an analogy for the industrial ecosystem.

After the gaps detected in the models and studies, the need to establish a Holonic proposal for the reengineering of manufacturing systems that strengthens the weaknesses of the current models has been identified and justified, making a series of contributions to achieve the incorporation of the aspects of sustainability, multi-level and multi-scale aligned with the objectives of the Agenda 2030 and based on digitisation. Likewise, the proposed Holonic architecture is based on the concepts of Industry 4.0 through the incorporation of CPS, proposing the HMS as an organisational enabler that manages the emerging complexity of current manufacturing systems, and that integrates and takes advantage of the benefits offered by digital and technological enablers. Proposal for the reengineering of sustainable cognitive manufacturing.

## 3. Proposal for the Reengineering of Sustainable Cognitive Manufacturing

Based on the review carried out in the previous section, corresponding to the different digital enablers at the operational levels of edge, fog and cloud, as well as the current

state of cognitive manufacturing, it allows to focus on the area of improvement that is the objective of this paper. In this section, the current limitations of cognitive manufacturing to meet the needs of sustainable manufacturing are analyzed, and a reengineering of cognitive manufacturing is proposed from the holonic organizational enabler and the support of enablers from Industry 4.0, multilevel (value chain, industrial plant, process, industrial equipment, etc.) and multiscale or location-independent.

Holonic paradigm is proposed in this work as a framework, in order to advance in the line of the possibilities offered by the concept of CPS, surrogate model and the requirements of integration of the principles of circular economy, the concept of metabolic rift, which refers to the reversal of the separation between the natural and the social, through the management of indicators or the integrated key performance indicators (KPI) of the economic, social and economic.

### 3.1. Holonic Framework for Conceiving Cognitive Manufacturing Systems from Cyber Physical Systems

Circular economy concept, as a paradigmatic framework for sustainability, is based on the following three fundamental principles [140]: preserve and improve the natural capital, optimize the resource performance, and enhance the effectiveness. Among the principles of the circular economy paradigm is the search for natural solutions, there is a tendency to adopt bioinspired solutions, in this sense the Holonic model inspired by natural systems is adopted.

Holonic systems, as bioinspired entities, are oriented from the three pillars of sustainability. These entities are articulated based on the following principles [141]: the reality consists of holarchies, holons are twofold entities, every holarchy matrix is fractal, multiscale and multilevel. The holon, as CPS driven by data in the three dimensions of sustainability, has three set point indicators or KPIs, which are oriented to continuous improvement through surrogate models.

The interest of holonic architecture lies in the fact that it is a minimal complexity fractal architecture, which constitutes a value to mitigate the static and dynamic complexity of manufacturing systems at different levels of granularity.

In the following sections, an ontological holonic architecture of the sustainable cognitive value chain and sustainable cognitive manufacturing systems is proposed, indicating the reengineering processes to be carried out for the mitigation of metabolic rift through the paradigm of the circular economy. This is specified in later sections with a greater degree of detail referring to the level of concretion of the industrial plant.

### 3.2. Holonic Reengineering of the Cognitive Value Chain

The digital transformation process described in Figure 2 has been carried out on the value chain, as a more aggregate manufacturing system. The objective is to achieve a cognitive and sustainable value chain 4.0 through the circular economy and holonic paradigms.

As indicated in Figure 2, the following steps have been followed:

(1) Digitization of the value chain and the transformation of the elements that integrate it as CPS, managing efficiency through quality or economic KPIs.

(2) Transition to environmental sustainability by integrating information on the environmental behaviour of the value chain, under some of the frameworks or paradigms such as the circular economy, through management of vectorial KPIs.

(3) Transition to sustainability under the triple bottom line concept, to mitigate metabolic rift by managing the three KPIs of the sustainability pillars in an integrated way to reverse metabolic rift.

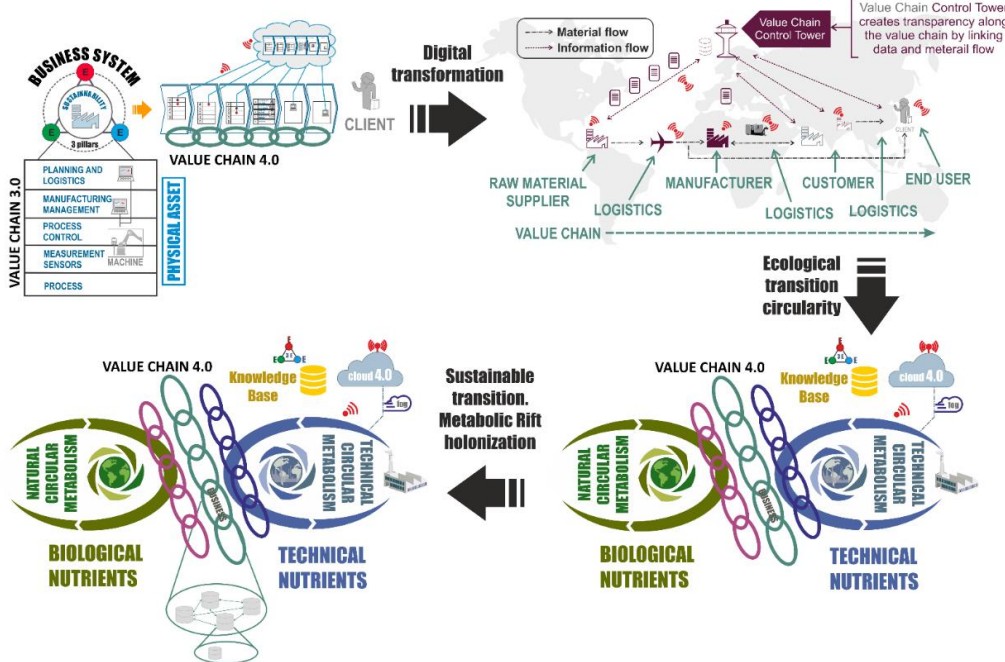

**Figure 2.** Digital transformation of the holonic value chain.

For the management of the transition from sustainability under the concept of metabolic rift, a holonic architecture showed in Figure 3, is proposed that makes it possible to analyze from the different views the efficiency, energy, water, materials, cyclicity, toxicity, etc. Likewise, in the domain of collaboration it works with different tools contained in the life cycle knowledge base.

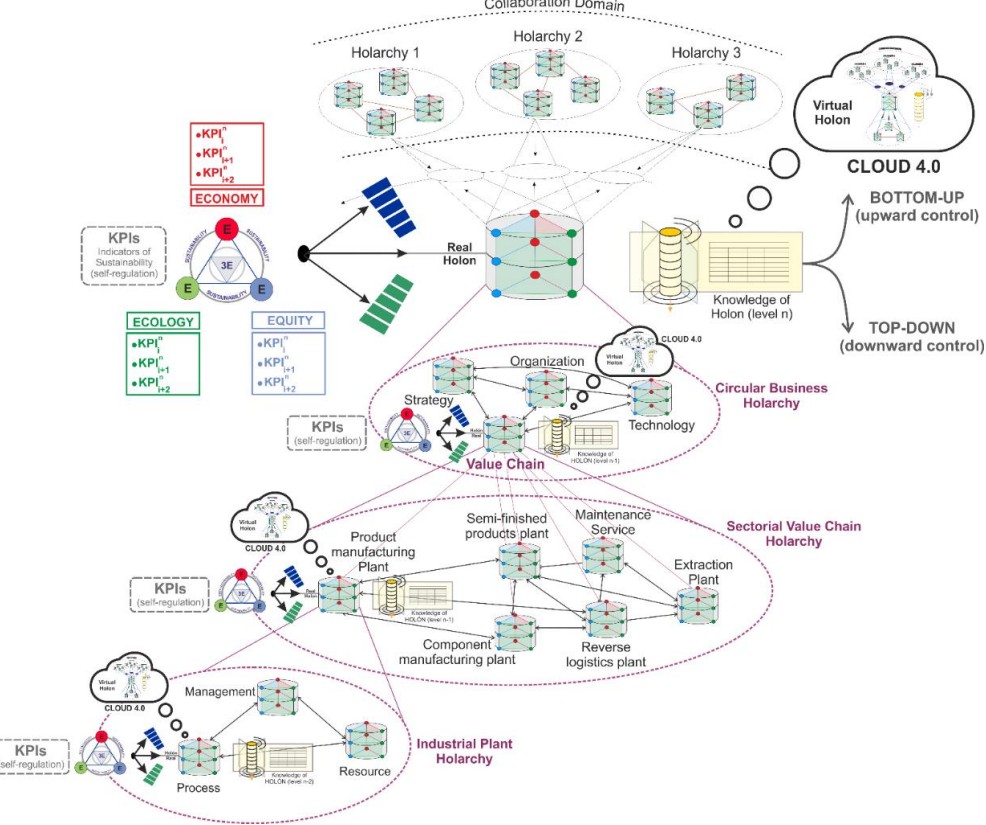

**Figure 3.** Holonic architecture of the cognitive and sustainable value chain.

In the knowledge base of the different holons, which corresponds to the levels of different granularity of the value chain, or geographical location for localized manufacturing systems, there is the knowledge base and cognitive computing strategies to obtain a cognitive and sustainable value chain.

*3.3. Holonic Reengineering of the Cognitive Manufacturing*

Based on the above, the reengineering process of cognitive manufacturing from the holonic paradigm allows:

(1) Address the goal of the sustainable smart factory by having a variety of management, operation and value creation closer to natural systems, with intelligence and knowledge by, for and based on action. Ecocompatibility is ensured based on embodied and situated intelligence, creating ecocompatible factories that mitigate and/or reverse metabolic rift.

(2) Incorporate embodied and situated learning from the continuous improvement of Lean teams into knowledge engineering processes, taking part of the organization's explicit and tacit know-how, making it possible to co-evolve with the market. This allows the embodied intelligence to be constituted through evolutionary surrogate models driven by data from the cloud.

(3) Mitigate the static complexity of manufacturing by being conceived under self-similarity criteria of cognitive CPS, which are self-optimizing, goal-oriented and self-organizing based on holonic entities of physical-bio-psycho-socio-cultural inspiration that determines a variety ecocompatible with nature.

(4) Consider operators as CPS, considering the cognitive dimension and their operational and learning styles. In this context, integrating operators and the human factor under cognitive CPS approach, it is possible to incorporate the augmented cognition paradigm, which incorporates the emerging knowledge of intelligence in technical and social environments.

In this way, the smart factory is constituted as a multilevel and multiscale (distributed) CPS ecosystem, giving rise to the holonic cognitive factory, as shown in Figure 4.

Figure 4 shows in the cooperation domain, the engineering architecture of a product design and development holon and, its manufacturing under the ISO 10303 standard and the process holarchies of Application Protocol AP238 (STEP-NC). This concretion opens up new possibilities for data communication between CAD/CAM and CNC manufacturing systems [142].

What is exposed in the above sections determines the need for a framework to conceive intelligent manufacturing systems as cognitive holons. Which is specified in having a design methodology [143] of cognitive connected intelligent CPS holons.

The architecture of this system in its informational view must consider the aspects of multilevel (micro, meso and macro), multiscale spatial (distributed manufacturing) and time in the operations, tactical and strategic dimension. In this sense, and based on the constraints of technology, it is necessary to start from an information system architecture on the (physical) edge, in the fog at the short-term holon interface, and a long-term information system or in the cloud linked to aspects of the business, as is shown in Figure 4. Both systems—cloud and fog—have different purposes and objectives oriented in the short and long term [144], thus such as the improvement of surrogate models of exploitation.

The definition of the cloud, fog and edge structure is determined by the latency time of the 4G or 5G communication infrastructure and the time requirements of the production system depending on the need or not for real-time execution.

Communication delay, in other words, the latency of the system is something that is currently important under the 4G model. In the future with 5G technology it will lose importance and enable real time control at the edge with surrogated models from cloud. A proposed infrastructure based on cloud, fog and Edge is provided in order to determine an acceptable latency time from edge to cloud, owing to the existence of Fog for real-time operation of Manufacturing Execution Systems (MES).

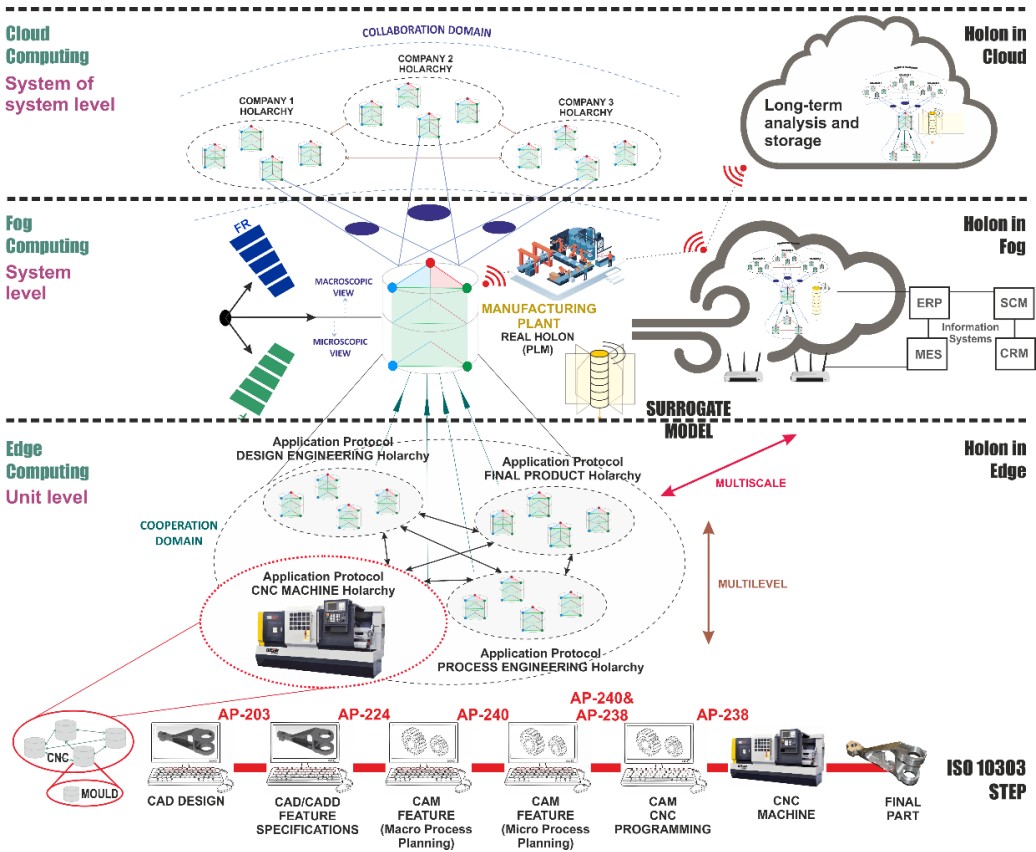

**Figure 4.** Architecture of holonic cognitive manufacturing and a concretion in STEP scenario for design and manufacture of machined parts.

### 3.4. Sustainable Holonic Cognitive Cyber-Physical System

CPS would constitute the basic holonic entity of the holonic cognitive factory. A cognitive CPS, like all CPS with embedded knowledge, would be formed by a physical part (robot, work table, etc.) that is analogous to the biological body of intelligent beings and a virtual part in the cloud and/or in the fog that supports the model of cognitive intelligence (mind), embedded in a physical body.

Industrial CPS is the first enabling technology for Industry 4.0, configured as an emerging data-driven enabler focused on creating manufacturing intelligence using ubiquitous networks with real-time data flows. These systems allow objects and processes, which are related in the physical world (robot, numerical control machine, etc.), to have a virtual representation in the cloud and in the fog, so that they are closely coupled and their efficiency is evaluated using predictive data analysis techniques [145] (e.g., machine learning model) and simulation models from the cyber world in the cloud.

Cloud-based CPS architectures are struggling to deploy because they do not meet the needs for decentralization, security, privacy, and reliability [146]. Latency times that take place between cyber-physical entities (smartphones, tablets, etc.) when operating on them in the virtual image and access to the cloud are too long, posing privacy and security problems, among others. The same occurs in obtaining real-time data flow for further processing by Big data, and analytical learning to make predictions of equipment failure or formulate surrogate models for operational efficiency control.

Nowadays, cloud computing and service orientation seems to be the right framework. However, cloud computing conflicts with Industry 4.0 when it comes to decentralized decision-making and reliable real-time control.

In this models sense, smart factory is articulated with a model as is shown in Figure 5, where there are CPS that use fog computing to implement machine learnings [60]. With this architecture, the holonic CPS can have three feedback loops, as shown in Figure 5.

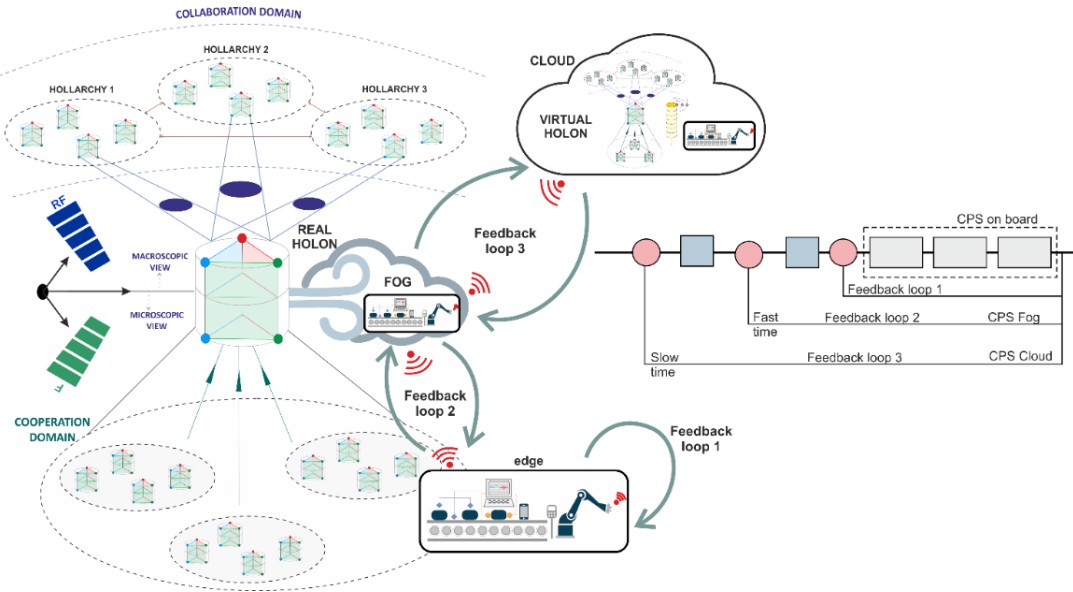

**Figure 5.** Holonic CPS feedback loops.

A first local loop on the edge, a second loop in the fog under surrogate intelligence models, and a third loop from the cloud with new models surrogated from learning and continuous improvement. Then, the design of a cognitive intelligent holonic CPS proposes the integration of:

- Three management elements that operate from the triple bottom line under the circular economy in management and operation.
- Operation oriented to continuous improvement and real-time learning.
- Manufacture of low static and dynamic complexity, establishing an architecture in self-similarity, self-optimized and goal-oriented, which generates the required variety with the environment (market), ensuring multilevel (granularity) and multiscale scalability.

Once defined the cognitive intelligent cyber-physical holon, the instruction of the holon and the holarchies that would constitute the cognitive factory are detailed below.

The holon is self-regulating from the principles of life, that is, taking nature as a model, mentor and measure [147], towards the circular economy and under the three pillars of sustainability and cradle to cradle (C2C). The objective of obtaining a connected intelligent cognitive metabolism is pursued.

A tactical holarchy is defined that includes the modules of the Manufacturing Execution System (MES) belonging to the collaboration domain with the planning and programming holarchies Enterprise Resource Planning (ERP), supplier management Supplier Relationship Management (SRM), cycle management Product Lifecycle Management (PLM), etc. In its domain of cooperation is cognitive manufacturing, as an operational level holarchy that includes workstations and industrial equipment. In all cases holons are conceived as CPS and implemented using mind-body agent technology. Based on this, there are agents like the robot agent that is made up of body and mind, and other agents like the MES agent that only has mind.

The formulation of the architecture for cognitive holonic manufacturing requires the achievement of the following steps:

(1) Formulation of the objectives from the principles of sustainable life (circular economy, C2C, sustainability, etc.).

(2)    Specify the requirements of the different levels of collaboration and views of the holons.

(3)    Formulate the different stages of the life cycle, views of the complexity and level of granularity, as well as the requirements of the level of cooperation.

(4)    Establishment of the different types of agents and holarchies of reactive, deliberative or cognitive type.

(5)    Strategy analysis in the top-down and bottom-up life cycle. Embodied learning strategy for fog and in the case of cloud instruction and cognitive learning strategy and self-optimization.

(6)    In addition to the aforementioned aspects, it is necessary to consider the transversal aspects of cybersecurity and blockchain in all their potential.

(7)    Continuous improvement through the learning of conceptual cognitive models (in cloud) and mental models (in fog) through the ingestion of data in cloud and fog or the results of simulation in cloud in the previous phases of design or new operating conditions through digital twins.

### 3.5. Technological Mapping of Holonic Cognitive Sustainable Architecture

The proposal carried out for the design of the CPS, conceived as cognitive holonic CPS in the field of the cognitive intelligent factory, allows firstly to use the holonic architecture to carry out the holonization of the cognitive intelligent factory. That is, to characterize in the holonic architecture each one of the components of the manufacturing system. Once the holonization has been carried out, it is necessary to carry out the technological mapping of the holonic architecture, that is, describe the way in which the holonic conceptualization is implemented, through the cognitive holonic CPS, as a real (physical) and virtual (cyber) part.

As Figure 6 shows, it is proceeded to design cognitive cyber-physical holons. In this way, starting from the manufacturing cell and the identification of its physical structure (body) and intelligence (mind), the holonization of the manufacturing cell is carried out.

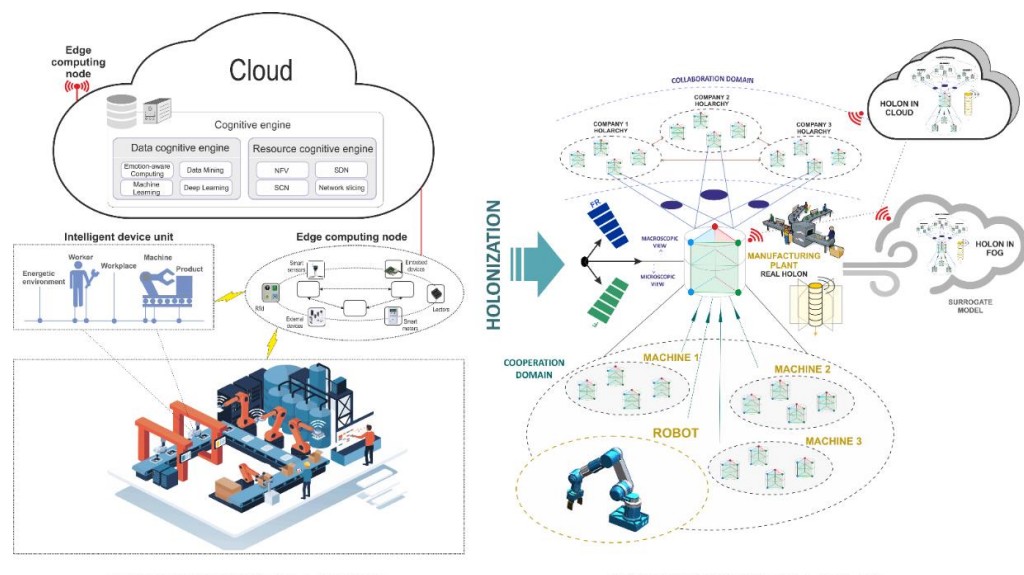

**Figure 6.** Manufacturing cell holonization as cognitive holonic CPS.

The informational view of the holon life includes, for each of the phases of the cycle, each of the stages necessary to constitute the informational view, as shown in Figure 7 adapted from [148].

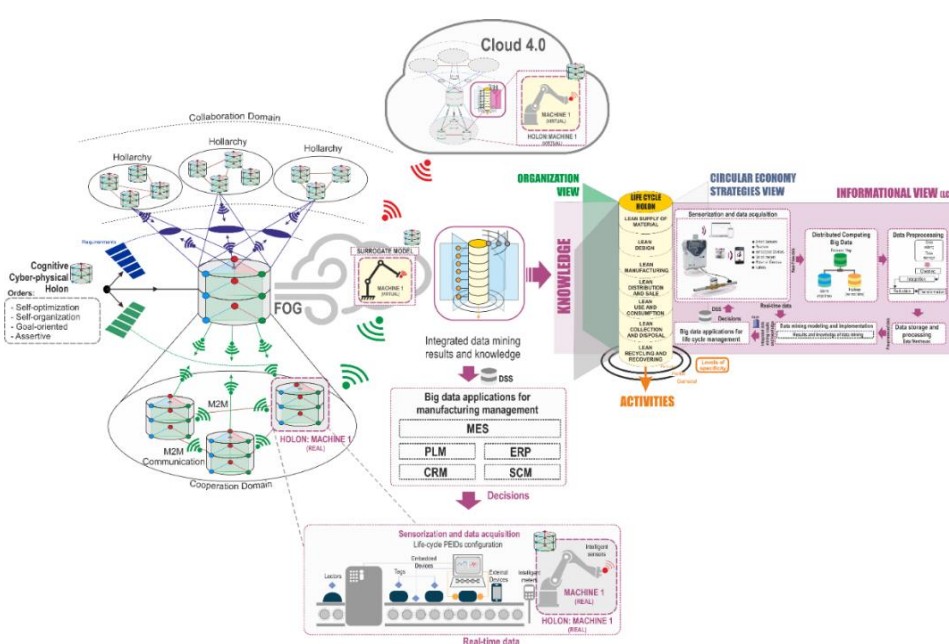

**Figure 7.** Technological mapping of the cognitive intelligent agent and informational view of the life cycle of the holon.

Firstly, it includes sensorization and data acquisition, followed by distributed computing in Big data, the preprocessing of data and its subsequent storage. Once the data capture and storage process is finished, this information is used in the realization, improvement and updating of the surrogate models that attend to the different levels of the production system of the value chain. Finally, these models and information are displayed and managed by the specific applications of each holon, at different stages of the life cycle, through indicators in integrated dashboards, among others.

The implementation of the informational view of the cognitive cyber-physical holon (holarchy) is carried out using intelligent agents, based on the descriptions carried out in this paper, transposed to the domain of intelligent agent technology through the cognitive intelligent agent as a cyber physical agent, as is described in Figure 7, adapted from [148].

Firstly, the edge detection layer contains the equipment and industrial systems to continuously acquire real-time measurements, where the cognitive intelligent agent is integrated to mediate communications between physical and cyber environments. At this level, the agent has the knowledge on board. Secondly, the fog layer contains technical components to receive incoming data streams, run models, and get results. At this level, the twin in the fog of the cognitive agent contains the parameterization of the model. Thirdly, the cloud platform stores production-ready machine learning models for different engineering applications and production phases, which are disseminated and executed by fog agents deployed within the facility's local network. Communications from the factory to the cloud depend on the facility's existing security policies and services that govern Internet communications.

Finally, once the cloud receives factory communications, a cloud database of registered devices is used to identify the engineering applications handled by the agent and return the relevant models to download or sync. The downloaded models are stored in the fog, so they can run within the physical limits of the factory and offer real-time predictions and decision making (for example, control changes) without persistent connections to the cloud.

The cognitive intelligent agent, in its virtual part, beyond its on-board knowledge, belongs to a collaborative domain in the cloud that could be called knowledge engineering (it collects data from the plant, from an external environment, etc., processes it with Big data and AI, make surrogate, cognitive models) and send them into the fog. It is in the fog the

place where they are parameterized (technological specification) for use according to the exploitation plan by the different departments, work stations, etc., according to the phase of the life cycle. In the use process, data is generated that in real time rises to the fog and cloud for the improvement of the surrogate models in a continuous improvement cycle.

### 3.5.1. Architecture Functional Specification Model

In terms of communication infrastructures, there is a very established tendency towards "network softwarization" in which once inflexible communications and services are being replaced by very flexible solutions that involve standard computing equipment, open standards and open source implementation [149].

The technological implementation of Holonic systems resulting from the reengineering of cognitive systems can be carried out using Open Manufacturing Platforms (OMP), or open technologies offered by different manufacturers such as BMW group, Siemens, IBM Watson Analytics or Microsoft Azure. An alternative ad hoc solution from the experience and information technologies that companies have available or wish to acquire. In both cases the specification of the Holonic architecture of a reengineering project in an architecture of microservices associated with cyber-physical holons is a tool or previous step to the implementation in a specific technology or manufacturing platform.

Container-based technology presents itself as an opportunity to implement the proposed holonic architecture. Specifically, the Arrowhead Framework [150] architecture is based on the use of containers, allowing the reduction of latency times, scalability, the integration of new components and increased security. To ensure interoperability in IoT, this architecture selects Service Oriented Architecture (SOA). SOA is characterized by a service-based data exchange between a service producing system and a service consuming system. In SOA, two systems do not need to know each other at the time of design to allow real-time data exchange. A new SOA service requires registering in the service registry where it can be detected by any other service on the network. Real-time coupling is initiated by an orchestration mechanism, primarily supported by the authentication and authorization mechanisms supported by an orchestration system that provides the requesting consumer with the endpoint of the selected producer. Each device and software system is responsible for its own data and functionality, and may be independent of other systems. Once a service exchange is established between two systems, this exchange can continue without the additional involvement of any supporting services or functionality.

Basic elements of the system that are mandatory to provide the minimum recommended services in a functional unit are described below and showed in Figure 8 [150]:

- Registration system: responsible for registering the services and enabling the discovery of registered services.
- Authorization system: is responsible for providing credentials to the systems in the functional unit, allowing system authentication and service exchange authorization. Manage access to specific resources using rules, control external access to specific resources, and publish authorizations.
- Orchestration system: it is responsible for providing information on service consumption patterns to the system registered in the local cloud. Manage connection rules for specific services. If necessary, the authorization system is consulted to verify whether the system consuming the service can be authenticated and authorized to consume the requested service.
- Gateway manager: helps locate services offered by neighboring containers. Located the service, it is responsible for establishing a secure communication route between client and server.

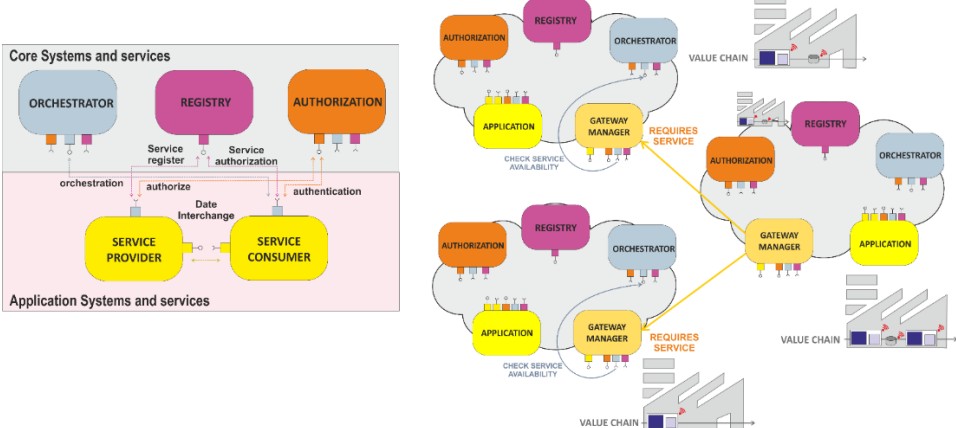

**Figure 8.** Basic holon microservices architecture and communication between functional units. Adapted from [150].

Once the basic elements that a functional unit must have are defined, the system applications (such as consumers and service producers) necessary to guarantee the functionality of the specific unit can be incorporated.

An essential requirement between functional units is the ability to communicate with each other. To do this, the gateway manager allows establishing secure communications between different functional units as showed in Figure 8. This connectivity between functional units allows the development of a multilevel architecture, providing units for monitoring and controlling a certain machine or units associated with the management of a process or even integrated into an architecture of a business management system or sustainable value chain.

### 3.5.2. Container-Based Technology Mapping

The implementation of technology can be on commercial platforms of manufacturing systems or with ad hoc technological mappings based on the digital transformation vision of the organisations. In this case, it has been chosen to illustrate the mapping of Holonic reengineering and its information model with microservices, based on container technology, due to scalability and security reasons and in order to overcome the resistance to migrate manufacturing information systems to the cloud.

The characteristics of container-based technology makes it appear as a suitable technology in which to map the proposed holonic architecture. In order to clarify the suitability of using this technology in the development of holonic system, the bijective relationship that exists between holonic paradigm and container-based technology is analyzed in Table 1.

When implementing the architecture in the technology presented, it should be noted that there are different container virtualization solutions based on free software in the market. These include LXC, OpenVz and Docker, with Docker presenting advantages in terms of flexibility, ease of use and integration. When it comes to container orchestrators, there is Docker Compose that allows static orchestration and Docker Swarm that enables dynamic orchestration (several servers). While there are other container orchestrators such as Kubernetes, developed by Google, or Marathon Mesos.

Once the basic elements of the functional unit have been defined, and the bijective relationship between holonic and container-based technology characterized, a mapping is made for the edge, fog and cloud, as shown in Figure 9.

**Table 1.** Bijective relationship between holonic and container-based architecture.

| Holonic Paradigm | Container-Based Technology |
|---|---|
| **Unit** | |
| Whole and Part. A holon can contain other holons. | A container can contain other containers. |
| **Relationship** | |
| Holarchy. Structures that refer to any bio-social organization with a certain degree of coherence, dynamic stability and harmony. Holarchy is formalized by holons as complete or dual basic entities, by relationships between holons as part of an entity and the rules and strategies of behavior that constitute the canon, along with the strategy of the whole holarchy. | Container relationships through the basic elements of the system (Orchestrator, Registration and Authorization) and interoperability (Gateway Manager). |
| **Levels** | |
| Multilevel. Refers to the aggregation of elements of the same or different scales, with specific derived properties of the scale to obtain an emergent property by aggregation such as, e.g. molecules, cluster of molecules, or microstructures, obtaining a compound with emergent properties different on a higher level. Therefore, level refers to the granularity of the analysis. | It is possible to model with containers from the level of sensors and actuators, at the manufacturing process level, to the implementation of the different modules of the management systems, at the management and planning level. It enables different degrees of granularity based on the different levels. |
| **Scale** | |
| Multiscale. It refers to the qualities or classes of interactions between elements that are given on the basis of a specific dimension or spatial scale of a given level, in which there are properties derived from the level (atomic, molecular, . . . , local, regional, global, etc.). | Scalability. It allows to create instances in different geographical locations, incorporating dynamic container orchestrators on different servers. |
| **Autonomy** | |
| The ability of a holon to create and control the execution of its plans and/or strategies, without the need for external assistance. | Each entity can act on its own independently of other systems, being responsible for its own data and functionalities. |
| **Structure** | |
| Self-similarity: It allows reducing the complexity of the systems, since the holons are homogeneous, having similar interfaces and behaviors, although their functions are differentiated. | Basic architecture. Each container has a fixed structure, made up of the basic elements of the system and system applications. |
| **Cooperation** | |
| It consists of the process by which a set of holonic entities develop mutually accepted plans and put them into practice. | The proper use of basic elements makes it possible to provide joint solutions and value-added services. |

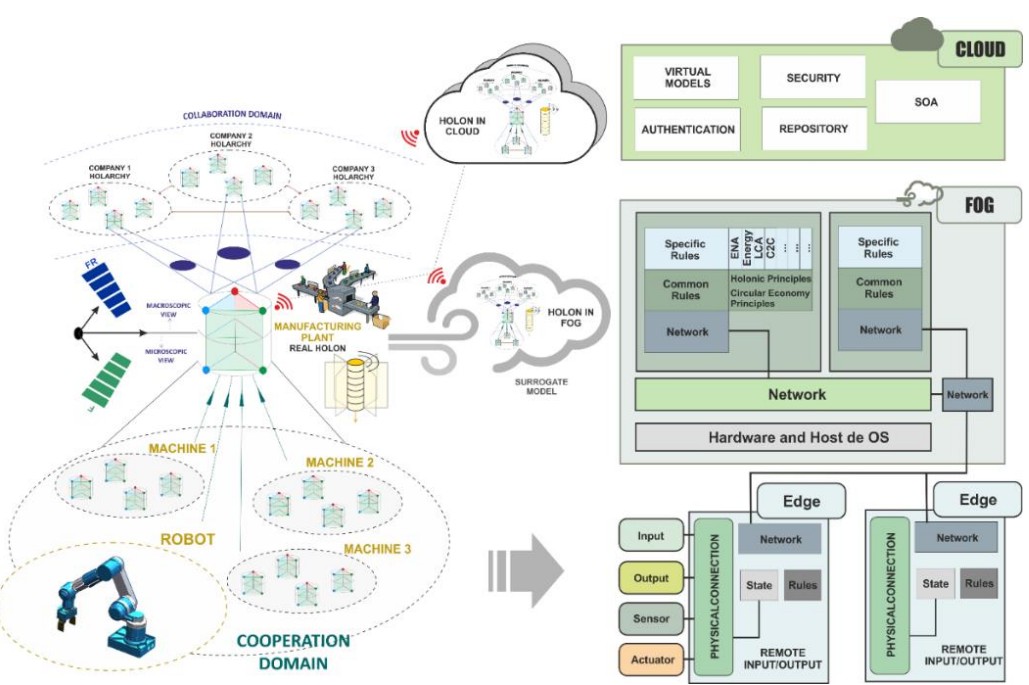

**Figure 9.** Container-based cognitive cyber-physical agent architecture.

## 4. Discussion

The reengineering processes in which small and medium-sized companies are concerned in the context of the possibilities offered by digital innovation centers can help to ensure that all companies, small or large, of high technology or not, can take advantage of digital opportunities. With technical universities or research organizations in the center, digital innovation acts as single windows where companies, especially SMEs, new companies, and medium-capitalized companies can get access to technology, financial advice, market intelligence and network creation opportunities.

Figure 10 shows a roadmap to carry out the reengineering of multilevel manufacturing systems (value chain, industrial plant, process and industrial facilities) and multiscale corresponding to the centralized and distributed manufacturing modes.

The roadmap establishes a tentative work process for the holonic reengineering team that is included in the sequence of steps that is included therein.

In the proposed roadmap, the following procedure is followed. Firstly, based on the conceptual model, it is established that the analysis will be carried out within the productive sector of the manufacturing industry, which has its own metabolism (industrial metabolism). Second, the circular value chain corresponding to the industrial sector is developed, which allows defining the entities involved. Next, the circular value chain is holonized through the holonic architecture proposed in this paper. For this, holons and holarchies of the collaboration and cooperation domains and their relationships are defined.

After the phase of the conceptual holonic domain, the informational domain is continued and the digital transformation of the circular value chain 3.0 to 4.0 is carried out, considering that the defined holons are cyber-physical holons. The developed holonic architecture is then implemented using intelligent agents. For this, the technological transfer of holons and holarchies to intelligent agents is carried out. Once the intelligent agents are defined, an ontology of knowledge is structured that, at the service of the holonic architecture, enables the exchange of information between the different agents that make up the system. The goal is to develop a computer application that models multi-level recursion that minimizes complexity in product design and development and associated manufacturing processes. Finally, the last phase corresponds to the prototype implementation process.

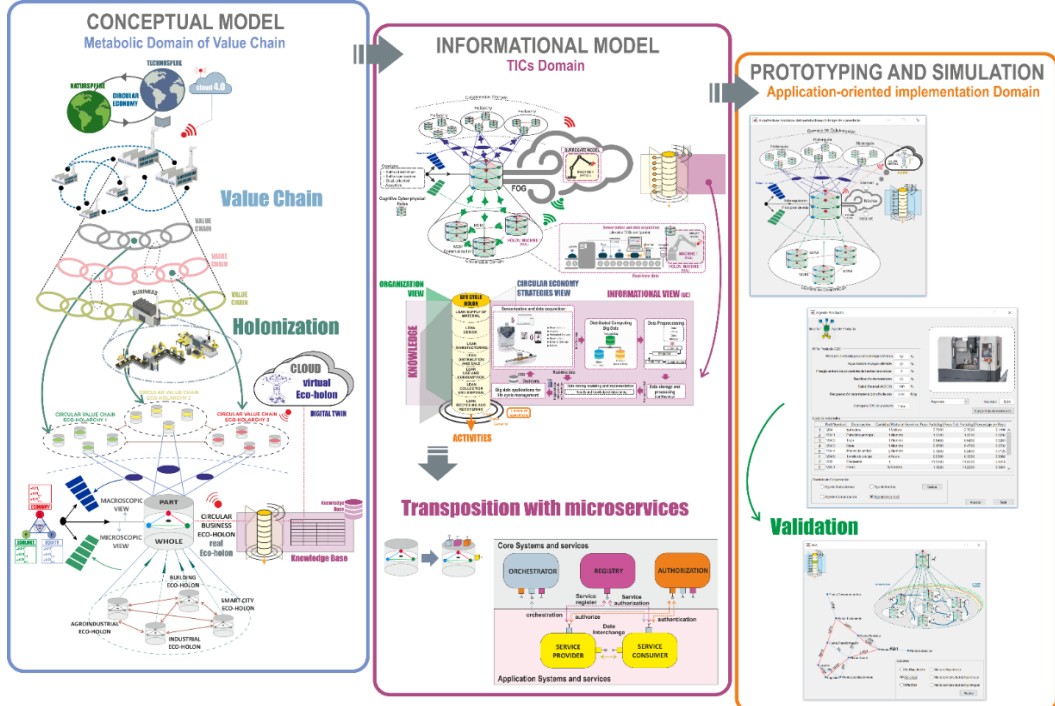

**Figure 10.** Roadmap for the holonic reengineering of cognitive cyber-physical systems for sustainable manufacturing.

An important aspect of the model that should be considered and addressed for future research work is cybersecurity. Cybersecurity should be considered when establishing the technology for implementing CPS through container technology. Cyber-attacks continue to develop at a very fast rate and with ever-increasing capability so it is important to establish the monitoring of the systems attending to three fundamental aspects: (1) Fake data injection is the hosting of malicious code in applications in order to attack those pages and/or collect user data. Some research suggests that this technique can be used to feed false information into the application database, remove important information or deny access to the owners or creators of the application database [151], (2) Analyzing the effect of replay attacks on constrained CPS to maximize the detection rate while keeping process degradation contained [152] and, (3) Denial of service, considered as an attack on a computer system or network, causes unavailability of a service or resource to legitimate users [153].

Although the proposed architecture is general for manufacturing systems, the research on CPS business models to improve Holonic architectures from lessons learned and successful cases from the commercial sector establishes another future line of study for implement the architecture using CPS Simatic of Siemens since Siemens Industrial Edge will allow the development of the proposed architecture based on Docker containers.

## 5. Conclusions

Some clear benefits are identified from the implementation of Industry 4.0. The most important benefits correspond to greater flexibility, quality standards, efficiency and productivity. This will allow massive customization of products, allowing companies to satisfy customer demands, creating value through the constant introduction of new products and services in the market. Furthermore, collaboration between machines and humans could have a social impact on the lives of the workers of the future, especially with regard to optimizing decision-making, guaranteeing their safety and integrity.

Based on the review carried out, it can be seen that the implementation of field enablers such as virtual and augmented reality, additive manufacturing, collaborative robotics, RFID, M2M or wearables, and cloud enablers such as big data, CMfg or IoT could be opportunities or threats for organizations. The fact that some technologies can result in both opportunities and threats is because the different areas are interconnected, with no well-defined limits between them, and depending on where it is analyzed, it could have a positive or negative connotation. Hence, the incorporation of an organizational enabler such as the holonic is justified to serve as a framework for the integration of Industry 4.0 technologies and to achieve the sustainable digitized value chain.

In this paper the reengineering of cognitive manufacturing is proposed from the perfect conjunction between the concept of CPS and the holonic paradigm. Therefore, the cyber-physical holon is developed through the informational view of the holon, where the holarchy of the informational view is defined to establish the different phases of the life cycle of the holon.

Finally, the technological mapping of the proposed holonic architecture based on the cyber-physical holon is developed, using intelligent agents, which allows the physical and virtual holon to be mapped as an intelligent agent at the edge level with a physical and virtual part (on-board knowledge), at the fog level for updating the surrogated models used by the agent, and at the cloud level as a digital twin where CPS exists as a parameterizable digital twin. All of the above opens up enormous possibilities for modeling the metabolism of manufacturing systems, from sensorization and data acquisition at the plant level, to parameterization and creation of models under continuous improvement tasks that optimize from efficiency, cyclicity and search of safety in technical and biological cycles.

**Author Contributions:** Conceptualization, A.M.-G., M.J.Á.-G. and F.A.-G.; Investigation, A.M.-G. and M.J.Á.-G.; Methodology, A.M.-G. and F.A.-G.; Writing-original draft, A.M.-G. and M.J.Á.-G.; Writing-review & editing, A.M.-G., M.J.Á.-G. and F.A.-G. All authors have read and agreed to the published version of the manuscript.

**Funding:** This research has received no external funding.

**Institutional Review Board Statement:** Not Applicable.

**Informed Consent Statement:** Not Applicable.

**Data Availability Statement:** Not Applicable.

**Conflicts of Interest:** The authors declare there to be no conflict of interest.

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
