# Peer review of "Holonic Reengineering to Foster Sustainable Cyber-Physical Systems Design in Cognitive Manufacturing"

_applsci, doi:10.3390/app11072941_

Round 1
Reviewer 1 Report
Dear Authors,
I reviewed the work, see in the attached file.
I wish you success!
Best regards,

Author Response
Dear reviewer,
Firstly, the authors would like to thank the reviewer for his/her work in reviewing the manuscript and for his/her suggestions on how to improve the paper. The new manuscript submitted includes modifications in light of his/her recommendations.
The language and formal aspects of the manuscript have been revised.
In this attached document, all comments will be answered with their reference in the text.
Kind regards!

Reviewer 2 Report
do all changes suggested, please, read the attached file

Author Response

(The authors gave the same response as above.)

Reviewer 3 Report
- Abbreviations must be explained the first time they appear in the text (line 44 SMEs)
- The introduction should be supported by references, for example, Line 46 to 48; line 49 to 50 "...value chain is identified as the metabolic rift..." reference; among others
- Review English line 98
- Line 254 - review title
- Line 533 - review English
- Line 533 to 558 - reference
- Line 585 to 586 - The last sentence is confusing
Author Response

(The authors gave the same response as above.)

Reviewer 4 Report
Comments to the authors
Manuscript ID: applsci-1127919
Title: Holonic reengineering to foster sustainable cyber-physical systems design in cognitive manufacturing
1) The contribution of the manuscript is unclear. Does it propose a new structure? Does it survey about available structures?
2) The manuscript studies cyber-physical systems. However, cyber attacks, as the inevitable threat to these systems, has not been discussed. The reviewer understands that cyber attacks might be out of the scope of the manuscript. However, the reviewer thinks that the authors should discuss this very important point, at least as a remark. Use the following articles to conduct the discussion; false data injection (10.1145/1952982.1952995), replay attack (10.1109/ALLERTON.2019.8919762), denial of service (10.1016/j.csi.2008.09.038).
3) Figure 5: what is the impact of communication delay on the architecture performance?
4) It is not clear what is the basis to determine the layer of the systems? How can one determine the implementation level of a system? Cloud? Fog?
5) Provide 1-2 future directions at the end of Section 5. This would help the interested readers to pursue this line of research.
Author Response

(The authors gave the same response as above.)

Round 2
Reviewer 2 Report
I think authors did not considered some of the previous comments and there were stuck with the old vision of holonic world. Since several years ago there are absolutely new approaches, in the 4.0, IoT, and etc. The examples of holonic instances were developed in the 90s and 2000s, they missed the important field of moulds, they missed the important fields of dies. Even after giving them some ideas, they disregard them. Even giving some examples, they did not check them. On the other hand, authors did not accomplish previous review.
Figures are nice but too general, even empty of meaning: please give real examples of holonic production today.
Holons are structures in cells, that were overwhelmed by the new lean manufacturing ideas. I really think that the paper is not new and the examples very weak (if some is included).
A total reflexion about topic and evolution would be recommended.
Author Response
Dear reviewer,
Firstly, the authors would like to thank again the reviewer for his/her work in reviewing the manuscript and for his/her suggestions on how to improve the paper.
The authors would like to highlight that the direction of our manuscript is towards this special issue called "Advances in Information and Communication Technologies (ICT)" and some of the suggestions indicated are oriented towards a very specific level of mechanical manufacturing which is not aligned with the aims of the manuscript. Nevertheless, the comments will be addressed in order to align them with the expected aims of the manuscript.
The new manuscript submitted includes modifications in light of his/her recommendations.
In the attached document, all comments will be answered with their reference in the text.
Yours faithfully,
María Jesús Ávila

Reviewer 4 Report
No more comments.
Author Response
Dear reviewer,
The authors are grateful that our modifications for the improvement of the paper were sufficient and accepted.
Yours faithfully,
María Jesús Ávila
Round 3
Reviewer 2 Report
Please, take your time, make a much better version, thing about new ideas adn make a complete new version.

Author Response
Dear reviewer,
Firstly, the authors would like to thank again the reviewer for his/her work in reviewing the manuscript and for his/her suggestions on how to improve the paper.
The new manuscript submitted includes modifications in light of his/her recommendations.
In this attached document, all comments will be answered with their reference in the text.
Yours faithfully,
María Jesús Ávila
